# Cancer Risk in Children and Young Adults (Offspring) Born after Medically Assisted Reproduction: A Systematic Review and Meta-Analysis

**Manuela Chiavarini [1]** , **Andrea Ostorero [2]**, **Giulia Naldini [2]** and **Roberto Fabiani [3,*]**

[1]  Department of Experimental Medicine, Section of Public Health, University of Perugia, 06123 Perugia, Italy; manuela.chiavarini@unipg.it
[2]  Department of Experimental Medicine, Section of Public Health, School of Hygiene and Preventive Medicine, University of Perugia, 06123 Perugia, Italy; ostoreroandrea@gmail.com (A.O.); giulia.naldini@studenti.unipg.it (G.N.)
[3]  Department of Chemistry, Biology and Biotechnology, University of Perugia, 06123 Perugia, Italy
*   Correspondence: roberto.fabiani@unipg.it

**Abstract:** Many studies have investigated the relationship between medically assisted reproduction (MAR) and health outcomes, particularly cancer, in the offspring. This meta-analysis investigated the association between MAR and childhood cancer. Data sources were PubMed, Scopus, and Web of Science up until June 2018. From the selected studies, we extracted the cancer risk estimates of the exposure of interest (MAR, assisted reproductive technology—ART, and in fitro fertilization—IVF). We conducted the meta-analysis using a random effects model. The outcomes of interest were childhood cancers, classified according to the international classification of childhood cancer (ICCC-3). In our meta-analysis (18 cohort and 15 case-control studies) the overall cancer risk was significantly increased in children conceived by MAR, ART, or IVF. MAR and ART significantly increased the risk for hematological tumors, hepatic tumors, and sarcomas (odds ratio (OR) 1.54; 95% CI 1.18–2.02 and OR 1.92; 95% CI 1.34–2.74, respectively). MAR increased acute myeloid leukemia risk (OR 1.41; 95% CI 1.02–1.95) and ART increased neural cancer risk (OR 1.21; 95% CI 1.01–1.46). Our results suggest an increased risk of cancer in children conceived by MAR. Further studies are needed to investigate the impact of fertility treatments, parental subfertility status, and their association on health outcomes in the offspring.

**Keywords:** cancer; child; assisted reproductive techniques; in vitro fertilization; meta-analyses

---

## 1. Introduction

Since the first child born after in vitro fertilization (IVF) in the United Kingdom in 1978, assisted reproductive technologies (ART) for treatment of fertility problems have been increasing. ART are defined as all interventions that include the in vitro handling of both human oocytes and sperm, or of embryos. for the purpose of reproduction, including IVF and intracytoplasmic sperm injection (ICSI). Up to now, approximately 8 million children have been born worldwide following ART. In Europe, about 3% of all births are a result of ART, accounting for an estimated 170,000 births each year [1–3].

Moreover, in this context, it is important to consider that in addition to ART, other types of fertility treatments have been used. Indeed, "The International Glossary on Infertility and Fertility Care, 2017" [4] described medically assisted reproduction (MAR) as the "reproduction brought about through various interventions, procedures, surgeries and technologies to treat different forms of fertility impairment and infertility".

Among children born following MAR, several studies observed an increased risk of adverse short-term birth outcomes, such as multiple births, preterm births, and congenital malformations [5–8].

While much has been reported about these outcomes, relatively few studies have focused on the potential long-term adverse health effects of MAR use [9]. Some evidence indicates that ART may be responsible for the increase in the risk of somatic morbidity during childhood [10].

The risk of somatic morbidity includes childhood cancers, which have a great relevance as they are the second most common cause of death in children in developed countries [11].

The etiology of childhood cancer remains largely unclear, but it has been hypothesized that some of them are initiated during the early stages of fetal development [12]. Since the events leading to and carried out around conception can play an important role in childhood cancer, MAR may be a factor risk for this disease.

In view of the rapid growth of the population that uses MAR to solve infertility problems, it is very important to continually monitor its possible long-term adverse health effects. A previous meta-analysis indicated a significantly increased risk of 33% for all cancers in children born after MAR [13]. Moreover, in a subset of children born after ART, the risk of cancer was increased by 40% [13]. Since then, several other studies with wider sample sizes and longer follow-up times have been published with contrasting results. Therefore, we conducted a systematic review and meta-analysis to summarize the evidence and to derive a more accurate estimation of cancer risk in offspring born after MAR.

## 2. Materials and Methods

We followed the standard procedures for conducting and reporting a meta-analysis as recommended by MOOSE (meta-analysis of observational studies in epidemiology) guidelines and the PRISMA (preferred reporting items for systematic reviews and meta-analyses) statement [14,15].

### 2.1. Search Strategy and Data Source

We carried out a comprehensive literature search, without restrictions, up to 3 July 2018, through PubMed (http://www.ncbi.nlm.nih.gov/pubmed/), Web of Science (https://www.webofknowledge.com/), and Scopus (https://www.scopus.com/) databases to identify all the original articles on the association between MAR and cancer in children and young adults. The medical subject headings (MeSH) and key words used for search are in Supplementary Table S1. To identify additional relevant publications, we manually examined the reference lists of included articles and recent relevant reviews.

We systematically reviewed and selected the studies meeting the following criteria of eligibility: (i) assessed MAR and/or ART and/or IVF; (ii) reported at least one case of the selected outcome; (iii) used a cohort or case-control study design; and (iv) reported a risk estimate (standardized incidence ratio—SIR, hazard ratio—HR, relative risk—RR, or odds ratio—OR) for cancer in children and young adults, as well as its 95% confidence interval.

For each potentially included study, two investigators (AO and MC) independently selected the studies, extracted the data, and performed the quality assessment. Disagreements were resolved by discussion and, if necessary, in consultation with a third author (GN). Although useful for background information, reviews and meta-analyses were not included. We did not exclude studies for weakness of design or data quality. From the selected studies, we extracted information about study characteristics (study name, authors, publication year, study design), study population characteristics, exposure assessment, type of MAR treatment, outcomes, and variables of adjustment. When multiple estimates were reported in the article, we extracted those adjusted for the most confounding factors.

We considered the possible overlapping results in papers reporting the same study period or study population. We noticed that in the study by Sundh et al. [16], the risk data partially overlap with the results of the studies by Hargreave et al. [17], by Petridou et al. [18], and by Reigstad et al. [19]. These studies were included in our meta-analysis as they provided more information (e.g., age) needed for the stratified analyses.

We grouped the fertility treatments and procedures as MAR and ART, according to "The International Glossary on Infertility and Fertility Care, 2017" [4]. MAR includes all types of fertility treatments, in particular any treatment inducing, triggering, stimulating ovulation, and any ART procedure, while ART includes all interventions that involve the in vitro handling of both human oocytes and sperm, or of embryos, for the purpose of reproduction, such as IVF and ICSI. We considered it correct to stratify for IVF and ICSI (as specified in the original articles) because they are the most common ART techniques [2]. Unfortunately, we found very few data on ICSI, so we considered only IVF in the stratified analysis.

The outcome of interest in our analysis was childhood cancer, classified according to the international classification of childhood cancer (ICCC-3) [20]. We conducted separate meta-analyses for different cancer outcomes and the main cancer outcomes were hematological cancers, neural tumors, neuroblastoma, retinoblastoma, renal tumors, hepatic tumors, bone tumors, soft tissue and other extraosseous sarcomas, and germ cell tumors. Any cancer not included in the eight previous categories was classified as other cancers. We further explored the association with overall and specific cancer outcomes, based on the available evidence. All analyses were conducted separately for all types of MAR, all types of ART, and IVF. Based on the Newcastle–Ottawa scale (NOS) method [21], the quality of the studies was assessed by a 9-star system; a total score of ≥7 was used to indicate a high-quality study. No study was excluded due to NOS criteria, then all studies went to sensitivity analysis.

*2.2. Statistical Analysis*

We evaluated the association between MAR and cancer in children and young adults using the statistical program ProMeta version 3.0 (IDo Statistics-Internovi, Cesena, Italy).

For the overall estimation, the hazard ratio was taken as an approximation to the OR, and the meta-analysis was performed as if all types of ratio were ORs. The combined risk estimate was calculated using a random effects model, in which the effect measures were SIR, HR, RR, or OR. We conducted the analysis considering MAR as the exposure factor. Then, we investigated the cancer risk related to ART and further stratified the analysis to estimate the cancer risk associated with IVF.

We assessed heterogeneity between studies by the Cochran's Q statistic ($\chi^2$), deeming $p < 0.05$ as significant, and $I^2$ test, which yields results ranging from 0% to 100% ($I^2 = 0\%$–25%, no heterogeneity; $I^2 = 25\%$–50%, moderate heterogeneity; $I^2 = 50\%$–75%, large heterogeneity; and $I^2 = 75\%$–100%, extreme heterogeneity) [22,23]. To explore the sources of heterogeneity among studies and test the robustness of the associations, we conducted subgroup analyses and several sensitivity analyses. We further examined the influence of individual studies on the overall risk estimate, which was investigated by recalculating the pooled estimates for the remainder of the studies by omitting one study at each turn.

Publication bias was evaluated using the methods by Begg and Mazumdar [24] and by Egger [25], which both assess funnel plot asymmetry, the former based on the rank correlation between the effect estimates and their sampling variances, and the latter on the basis of a linear regression of a standard normal deviate on its precision. If the intercept of Egger's regression line deviated from zero with a *p*-value <0.10, the funnel plot was considered asymmetrical. In case of a small number (25 or fewer) of studies enrolled in the meta-analysis, as in the current review, this test for asymmetry possesses relatively low power to detect a real publication bias. If a potential bias was detected, sensitivity analyses were performed to assess the robustness of our findings. *p*-values reported are from 2-sided statistical tests and differences with $p < 0.05$ were considered significant.

## 3. Results

*3.1. Study Selection*

From the primary literature research through PubMed (*n* = 501), Web of Science (*n* = 559), and Scopus (*n* = 795) databases, we obtained a total of 1,855 records. After removing duplicates (*n* = 741), we identified 1114 records for title and abstract revision (Figure 1). Among these, we excluded 1050 articles

due to them not investigating the association between MAR and the outcomes of interest. Sixty-four studies were subjected to full-text revision. Examining the reference lists of both selected articles and recent relevant reviews, four other potentially eligible articles were identified. Subsequently, 36 papers without risk estimation were excluded. Therefore, at the end of the selection process, 32 studies [16–19,26–51] were enclosed in the systematic review and meta-analysis. In the study by Petridou et al. [18], there were two sets (Sweden and Greece) with different population and results, which we considered as two separate studies (Petridou-a, Petridou-b).

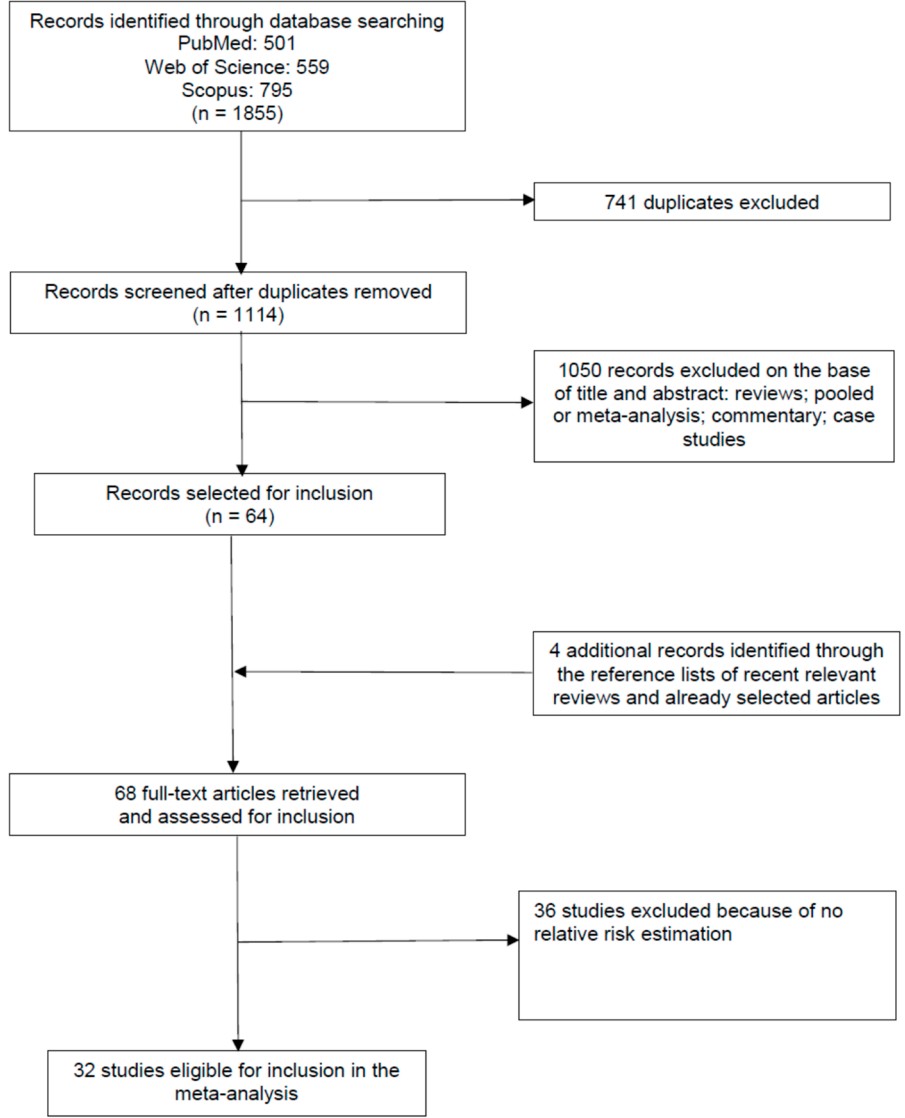

**Figure 1.** Flowchart of Study Selection.

### 3.2. Study Characteristics

Of the 33 selected papers, 15 were case control published between 1996 and 2014, while 18 were cohort studies, published between 1998 and 2018. One third of all papers were published in the last six years, and the most of them were cohort studies (Supplementary Table S2).

In our selection, the risk estimation was reported as OR for 15 studies [18,29,33,35,38–41,52], RR for six studies [19,31,32,36,46,51], SIR for four studies [27,30,43,50], and HR for seven studies [17,44,47–49,53,54].

Five studies reported tumors occurring in children aged under six years [27,32,38,40,41], twenty-two studies reported tumors occurring in children aged <15 years [16,18,19,28,30–32,34–36,38–43,45,50–52], while the remaining eleven studies reported the risk estimation in subjects with different ages (<35 years) [17,26,28,33,37,44,47–49,51,53].

Due to the complexity of the data reported in the 33 included studies, we synthetized the 20 different types of MAR and the 47 different outcomes in the Supplementary Table S3.

### 3.3. Quality Assessment

Study-specific quality scores are summarized in last right column of Supplementary Table S2. The quality scores varied in the range from five to nine (median: seven; median for cohort studies: eight; median for case control studies: six). High-quality studies (i.e., those studies that had a score greater than or equal to seven) included 17 cohort studies [16,17,19,27,30–32,37,43,44,46–51,53] and seven case-control studies [18,33–35,38,42,45].

### 3.4. Meta-Analysis

In consideration of the multiplicity of the outcomes of interest, we organized the results based on the ICCC-3 "International Classification of Childhood Cancer, Third Edition" by Eva Steliarova-Foucher [20]. After the overall analysis, we investigated all outcomes that were significantly associated with MAR, exploring the potential relation with any fertility treatment (MAR, ART, and IVF), and stratified for age and study design.

Due to the complexity of the meta-analysis, no forest plots will be shown, however they are available on request.

### 3.5. Overall Analysis

The overall analysis showed that MAR, ART, and IVF significantly increased the risk of cancer by 21%, 34%, and 41%, respectively (Table 1). After stratification for study design, we observed a significantly increased risk both in case-control and cohort studies.

In case of IVF, the association in case-control studies was at the limit of significance (OR 1.28, 95% CI 1.00–1.63). Stratifying for age, MAR and ART significantly increased cancer risk in the "<15 years" and "all others" groups, whereas no significant association was found in children aged under six years. Moreover, IVF significantly increased cancer risk in the "<15 years", while no further analyses were performed for "<6 years" and "all others" categories, due to the paucity of records.

In the overall analysis, the heterogeneity was significantly moderate for MAR ($I^2$ = 36.61%, $p < 0.0001$) and for ART ($I^2$ = 41.38%, $p < 0.0001$), while it was significantly high for IVF only in cohort studies ($I^2$ = 56.01%, $p = 0.045$). In the stratified analysis, the heterogeneity was significantly high for MAR in the "<6 years" ($I^2$ = 50.20%, $p = 0.029$).

Considering the studies that did not specify the cancer sites (referred to as "all cancer"), MAR and ART showed a significant increase of cancer risk by 9% and 11%, respectively (Table 1). IVF showed no statistically significant association with "all cancer". Stratifying for age, we found a significant increase of "all cancer" risk associated to MAR (13%) and to ART (22%) in the "all others" category (Table 1).

**Table 1.** Results of stratified analysis on the overall cancer risk estimates in children born after medically assisted reproduction (MAR), assisted reproductive technologies (ART), and in vitro fertilization (IVF) on the basis of study design and age.

| Models | Studies | Effect Size | Combined Risk Estimate | | Test of Heterogeneity | | | Publication Bias | |
|---|---|---|---|---|---|---|---|---|---|
| **MAR** | N° | N° | Value (95% CI) | *p* | Q | I²% | *p* | *p* (Egger Test) | *p* (Begg Test) |
| **All cancer sites** | 33 | 218 | 1.21 (1.14–1.28) | <0.0001 | 342.30 | 36.61 | <0.0001 | 0.101 | 0.897 |
| **Study design** | | | | | | | | | |
| Case-control | 15 | 73 | 1.16 (1.04–1.30) | 0.010 | 100.92 | 28.66 | 0.014 | 0.070 | 0.329 |
| Cohort | 18 | 145 | 1.22 (1.14–1.31) | <0.0001 | 240.80 | 40.20 | <0.0001 | 0.019 | 0.603 |
| Cohort excluding William et al., 2013 | 17 | 128 | 1.20 (1.12–1.28) | <0.0001 | 192.02 | 33.86 | <0.0001 | 0.098 | 0.993 |
| **Age** | | | | | | | | | |
| <6 years | 5 | 11 | 1.06 (0.74–1.52) | 0.735 | 20.08 | 50.20 | 0.029 | 0.844 | 0.436 |
| <15 years | 22 | 128 | 1.20 (1.09–1.32) | 0.0001 | 235.80 | 46.16 | <0.0001 | 0.336 | 0.699 |
| All others | 11 | 90 | 1.20 (1.13–1.28) | <0.0001 | 106.50 | 16.43 | 0.100 | 0.108 | 0.840 |
| **All cancers (not specified)** | 16 | 43 | 1.09 (1.03–1.16) | 0.005 | 47.27 | 11.15 | 0.266 | 0.771 | 0.746 |
| **Study design** | | | | | | | | | |
| Case-control | 0 | 0 | - | - | - | - | - | - | - |
| Cohort | 16 | 43 | 1.09 (1.03–1.16) | 0.005 | 47.27 | 11.15 | 0.266 | 0.771 | 0.746 |
| **Age** | | | | | | | | | |
| <6 years | 1 | 3 | - | - | - | - | - | - | - |
| <15 years | 9 | 14 | 1.00 (0.90–1.11) | 0.989 | 10.55 | 0.00 | 0.649 | 0.182 | 0.324 |
| All others | 8 | 29 | 1.13 (1.05–1.23) | 0.001 | 33.32 | 15.97 | 0.224 | 0.505 | 0.896 |
| **ART** | | | | | | | | | |
| **All cancer sites** | 20 | 95 | 1.34 (1.22–1.47) | <0.0001 | 160.36 | 41.38 | <0.0001 | 0.003 | 0.293 |
| **Study design** | | | | | | | | | |
| Case-control | 6 | 13 | 1.29 (1.02–1.63) | 0.036 | 9.46 | 0.00 | 0.663 | 0.490 | 0.180 |
| Cohort | 14 | 82 | 1.35 (1.22–1.50) | <0.0001 | 150.71 | 46.25 | <0.0001 | 0.003 | 0.203 |
| **Age** | | | | | | | | | |
| <6 years | 2 | 2 | 1.40 (0.78–2.50) | 0.258 | 0.01 | 0.00 | 0.934 | - | - |
| <15 years | 15 | 72 | 1.30 (1.16–1.47) | <0.0001 | 132.67 | 46.48 | <0.0001 | 0.009 | 0.201 |
| All others | 5 | 23 | 1.38 (1.21–1.56) | <0.0001 | 24.46 | 10.05 | 0.324 | 0.040 | 0.895 |
| All others excluding Wennerholm et al., 2014 | 4 | 21 | 1.44 (1.25–1.66) | <0.0001 | 19.92 | 0.00 | 0.463 | 0.240 | 0.904 |
| **All cancers (not specified)** | 13 | 15 | 1.11 (1.02–1.20) | 0.015 | 11.76 | 0.00 | 0.625 | 0.524 | 0.520 |
| **Study design** | | | | | | | | | |
| Case-control | 0 | 0 | - | - | - | - | - | - | - |
| Cohort | 13 | 15 | 1.11 (1.02–1.20) | 0.015 | 11.76 | 0.00 | 0.625 | 0.524 | 0.520 |
| **Age** | | | | | | | | | |
| <6 years | 0 | 0 | - | - | - | - | - | - | - |
| <15 years | 8 | 10 | 1.03 (0.92–1.14) | 0.623 | 4.97 | 0.00 | 0.837 | 0.927 | 0.929 |
| All others | 5 | 5 | 1.22 (1.00–1.38) | 0.002 | 2.45 | 0.00 | 0.654 | 0.058 | 0.142 |
| **IVF** | | | | | | | | | |
| **All cancer sites** | 10 | 18 | 1.41 (1.11–1.78) | 0.005 | 22.57 | 24.67 | 0.164 | 0.963 | 0.570 |

**Table 1.** *Cont.*

| Models | Studies | Effect Size | Combined Risk Estimate | | Test of Heterogeneity | | | Publication Bias | |
|---|---|---|---|---|---|---|---|---|---|
| **MAR** | **N°** | **N°** | **Value (95% CI)** | *p* | **Q** | **I²%** | *p* | *p* **(Egger Test)** | *p* **(Begg Test)** |
| **Study design** | | | | | | | | | |
| Case-control | 5 | 12 | 1.28 (1.00–1.63) | 0.051 | 9.39 | 0.00 | 0.586 | 0.493 | 0.170 |
| Cohort | 5 | 6 | 1.80 (1.05–3.09) | 0.031 | 11.37 | 56.01 | 0.045 | 0.530 | 0.573 |
| **Age** | | | | | | | | | |
| <6 years | 1 | 1 | - | - | - | - | - | - | - |
| <15 years | 9 | 17 | 1.37 (1.07–1.76) | 0.012 | 21.93 | 27.05 | 0.145 | 0.994 | 0.742 |
| All others | 1 | 1 | - | - | - | - | - | - | - |
| **All cancers (not specified)** | 3 | 3 | 1.16 (0.68–1.98) | 0.598 | 2.61 | 23.47 | 0.271 | 0.923 | 0.602 |
| **Study design** | | | | | | | | | |
| Case-control | - | - | - | - | - | - | - | - | - |
| Cohort | 3 | 3 | 1.16 (0.68–1.98) | 0.598 | 2.61 | 23.47 | 0.271 | 0.923 | 0.602 |
| **Age** | | | | | | | | | |
| <6 years | - | - | - | - | - | - | - | - | - |
| <15 years | 2 | 2 | 0.87 (0.49–1.54) | 0.632 | 0.002 | 0.00 | 0.881 | - | - |
| All others | 1 | 1 | - | - | - | - | - | - | - |

### 3.5.1. Hematological Tumors

As mentioned above, the hematological tumors group included leukemias and lymphomas. MAR and ART significantly increased the risk of hematological tumors by 21% and 30%, respectively (Table 2). After stratification by study design, a similar effect was observed in cohort studies only. Stratifying by age, MAR significantly increased the risk by 50% in the "all others" group. No significant association was found between hematological tumors and IVF.

Considering "leukemias", MAR and ART increased the risk by 17% and 27%, respectively. The stratification by study design and by age resulted in a higher risk of leukemia associated with MAR in cohort studies (29%) and the "all others" group (61%).

Focusing on "ALL and "AML", MAR showed no significant association with the former while it significantly increased the latter by 41%. After stratification by age, MAR was associated with a higher risk of AML in the "all others" group (115%).

No significant association was found for ART or IVF with both ALL and AML.

No fertility treatment was significantly associated with "lymphomas".

In the analyses for hematological tumors, the heterogeneity was significantly moderate for MAR ($I^2$ = 42.77%, *p* = 0.0001) and for ART ($I^2$ = 36.31%, *p* = 0.037), and significantly high for ART in cohort studies ($I^2$ = 50.50%, *p* = 0.013). Considering leukemia, the heterogeneity was significantly moderate for MAR ($I^2$ = 47.85%, *p* < 0.0001) and for ART ($I^2$ = 40.34%, *p* = 0.033), and significantly high for MAR in cohort studies ($I^2$ = 56.11%, *p* = 0.012) and for ART in cohort studies ($I^2$ = 59.71%, *p* = 0.008). Instead, considering ALL, the heterogeneity was significantly high for MAR ($I^2$ = 57.43%, *p* = 0.001), for MAR in case-control studies ($I^2$ = 62.42%, *p* = 0.0004) and for MAR in "<15 years" group ($I^2$ = 62.09%, *p* = 0.001).

**Table 2.** Results of stratified analysis on the hematological tumors risk estimates in children born after MAR, ART, and IVF on the basis of study design and age.

| Models | Studies | Effect Size | Combined Risk Estimate | | Test of Heterogeneity | | | Publication Bias | |
|---|---|---|---|---|---|---|---|---|---|
| **MAR** | N° | N° | Value (95% CI) | p | Q | I²% | p | p (Egger Test) | p (Begg Test) |
| Hematological Tumors | 16 | 70 | 1.21 (1.07–1.36) | 0.002 | 120.57 | 42.77 | 0.0001 | 0.541 | 0.883 |
| -Hematopoietic | 1 | 3 | - | - | - | - | - | - | - |
| -Lymphatic and hematopoietic tissue | 1 | 1 | | - | - | - | - | - | - |
| Leukemias (ICCC-3 I) | 14 | 59 | 1.17 (1.03–1.34) | 0.018 | 111.22 | 47.85 | <0.0001 | 0.263 | 0.534 |
| -Leukemia | 14 | 26 | 1.13 (0.96–1.34) | 0.135 | 47.16 | 46.99 | 0.005 | 0.208 | 0.567 |
| -Other leukemia | 1 | 1 | - | - | - | - | - | - | - |
| -non-ALL | 2 | 2 | 1.34 (0.38–4.69) | 0.651 | 0.65 | 0.00 | 0.420 | - | - |
| -ALL | 7 | 18 | 1.07 (0.82–1.40) | 0.632 | 39.93 | 57.43 | 0.001 | 0.096 | 0.088 |
| -AML | 5 | 12 | 1.41 (1.02–1.95) | 0.039 | 10.22 | 0.00 | 0.511 | 0.693 | 0.583 |
| Lymphomas (ICCC-3 II) | 5 | 7 | 1.22 (0.88–1.67) | 0.232 | 7.04 | 14.83 | 0.317 | 0.203 | 0.024 |
| -Lymphoma | 4 | 4 | 1.13 (0.84–1.53) | 0.412 | 2.96 | 0.00 | 0.397 | 0.249 | 0.042 |
| -Lymphoma NH | 2 | 2 | 0.91 (0.44–1.88) | 0.802 | 0.01 | 0.00 | 0.930 | - | - |
| -Lymphoma Hodgkins | 1 | 1 | - | - | - | - | - | - | - |
| **Hematological** | 16 | 70 | 1.21 (1.07–1.36) | 0.002 | 120.57 | 42.77 | 0.0001 | 0.541 | 0.883 |
| **Study design** | | | | | | | | | |
| Case-control | 7 | 49 | 1.12 (0.95–1.32) | 0.164 | 88.93 | 46.03 | 0.0003 | 0.072 | 0.648 |
| Cohort | 9 | 21 | 1.32 (1.11–1.56) | 0.001 | 31.39 | 36.28 | 0.050 | 0.128 | 0.415 |
| **Age** | | | | | | | | | |
| <6 years | 1 | 3 | - | - | - | - | - | - | - |
| <15 years | 11 | 51 | 1.09 (0.94–1.26) | 0.255 | 91.57 | 45.39 | 0.0003 | 0.097 | 0.559 |
| All others | 5 | 19 | 1.50 (1.25–1.81) | <0.0001 | 22.90 | 21.41 | 0.194 | 0.038 | 0.074 |
| All others excluding Hargreave et al., 2013 | 4 | 16 | 1.77 (1.40–2.25) | <0.0001 | 15.95 | 5.96 | 0.385 | 0.466 | 0.242 |
| **Leukemias (ICCC-3 I)** | 16 | 59 | 1.17 (1.03–1.34) | 0.018 | 111.22 | 47.85 | <0.0001 | 0.263 | 0.534 |
| **Study design** | | | | | | | | | |
| Case-control | 8 | 48 | 1.13 (0.96–1.33) | 0.156 | 88.39 | 46.83 | 0.0002 | 0.078 | 0.625 |
| Cohort | 8 | 11 | 1.29 (1.01–1.64) | 0.038 | 22.78 | 56.11 | 0.012 | 0.574 | 0.876 |
| **Age** | | | | | | | | | |
| <6 years | 1 | 3 | - | - | - | - | - | - | - |
| <15 years | 12 | 48 | 1.08 (0.93–1.26) | 0.300 | 89.25 | 47.34 | 0.0002 | 0.087 | 0.516 |
| All others | 4 | 11 | 1.61 (1.22–2.14) | 0.001 | 13.31 | 38.71 | 0.091 | 0.167 | 0.186 |
| **ALL** | 7 | 18 | 1.07 (0.82–1.40) | 0.632 | 39.93 | 57.43 | 0.001 | 0.096 | 0.088 |
| **Study design** | | | | | | | | | |
| Case-control | 5 | 16 | 1.04 (0.76–1.42) | 0.829 | 39.92 | 62.42 | 0.0004 | 0.118 | 0.150 |
| Cohort | 2 | 2 | 1.18 (0.79–1.78) | 0.415 | 0.01 | 0.00 | 0.937 | | |
| **Age** | | | | | | | | | |
| <6 years | 1 | 1 | - | - | - | - | - | - | - |
| <15 years | 5 | 16 | 1.06 (0.79–1.44) | 0.693 | 39.57 | 62.09 | 0.001 | 0.129 | 0.150 |
| All others | 2 | 2 | 1.06 (0.62–1.81) | 0.836 | 0.21 | 0.00 | 0.646 | - | - |
| **AML** | 5 | 12 | 1.41 (1.02–1.95) | 0.039 | 10.22 | 0.00 | 0.511 | 0.693 | 0.583 |
| **Study design** | | | | | | | | | |
| Case-control | 4 | 11 | 1.29 (0.91–1.82) | 0.149 | 9.23 | 0.00 | 0.606 | 0.848 | 0.938 |
| Cohort | 1 | 1 | - | - | - | - | - | - | - |
| **Age** | | | | | | | | | |
| <6 years | 1 | 1 | - | - | - | - | - | - | - |
| <15 years | 3 | 9 | 1.18 (0.80–1.74) | 0.393 | 6.29 | 0.00 | 0.617 | 0.634 | 0.677 |
| All others | 2 | 3 | 2.15 (1.18–3.93) | 0.012 | 1.24 | 0.00 | 0.538 | 0.462 | 0.602 |
| **Lymphomas (ICCC-3 II)** | 5 | 7 | 1.22 (0.88–1.67) | 0.232 | 7.04 | 14.83 | 0.317 | 0.203 | 0.024 |
| **Study design** | | | | | | | | | |

**Table 2.** *Cont.*

| Models | Studies | Effect Size | Combined Risk Estimate | | Test of Heterogeneity | | | Publication Bias | |
|---|---|---|---|---|---|---|---|---|---|
| **MAR** | N° | N° | Value (95% CI) | *p* | Q | I²% | *p* | *p* (Egger Test) | *p* (Begg Test) |
| Case-control | 1 | 1 | - | - | - | - | - | - | - |
| Cohort | 4 | 6 | 1.32 (0.90–1.94) | 0.157 | 6.56 | 23.73 | 0.256 | 0.187 | 0.091 |
| **Age** | | | | | | | | | |
| <6 years | 0 | 0 | - | - | - | - | - | - | - |
| <15 years | 3 | 3 | 1.09 (0.67–1.80) | 0.724 | 2.26 | 11.34 | 0.324 | 0.197 | 0.117 |
| All others | 2 | 4 | 1.46 (0.82–2.58) | 0.199 | 4.63 | 35.23 | 0.201 | 0.347 | 0.497 |
| **ART** | | | | | | | | | |
| Hematological Tumors | 11 | 25 | 1.30 (1.08–1.58) | 0.006 | 37.68 | 36.31 | 0.037 | 0.234 | 0.726 |
| Leukemias (ICCC-3 I) | 11 | 20 | 1.27 (1.03–1.56) | 0.025 | 31.85 | 40.34 | 0.033 | 0.453 | 0.820 |
| -Leukemia | 9 | 10 | 1.10 (0.90–1.34) | 0.365 | 10.66 | 15.60 | 0.299 | 0.849 | 0.531 |
| -Other leukemia | 1 | 1 | - | - | - | - | - | - | - |
| -non-ALL | 1 | 2 | - | - | - | - | - | - | - |
| -ALL | 5 | 6 | 1.25 (0.94–1.64) | 0.123 | 4.11 | 0.00 | 0.534 | 0.407 | 0.188 |
| -AML | 1 | 1 | - | - | - | - | - | - | - |
| Lymphomas (ICCC-3 II) | 3 | 5 | 1.58 (0.94–2.66) | 0.083 | 4.97 | 19.55 | 0.290 | 0.457 | 0.327 |
| -Lymphoma | 3 | 3 | 1.36 (0.78–2.37) | 0.278 | 2.39 | 16.19 | 0.303 | 0.071 | 0.117 |
| -Lymphoma NH | 1 | 1 | - | - | - | - | - | - | - |
| -Lymphoma Hodgkins | 1 | 1 | - | - | - | - | - | - | - |
| **Hematological** | 10 | 25 | 1.30 (1.08–1.58) | 0.006 | 37.68 | 36.31 | 0.037 | 0.234 | 0.726 |
| **Study design** | | | | | | | | | |
| Case-control | 3 | 10 | 1.26 (0.96–1.65) | 0.095 | 9.36 | 3.81 | 0.405 | 0.513 | 0.245 |
| Cohort | 7 | 15 | 1.37 (1.05–1.77) | 0.019 | 28.28 | 50.50 | 0.013 | 0.151 | 0.553 |
| **Age** | | | | | | | | | |
| <6 years | 0 | 0 | - | - | - | - | - | - | - |
| <15 years | 9 | 18 | 1.09 (0.94–1.27) | 0.240 | 15.54 | 0.00 | 0.556 | 0.658 | 0.850 |
| All others | 1 | 7 | - | - | - | - | - | - | - |
| **Leukemias (ICCC-3 I)** | 11 | 20 | 1.27 (1.03–1.56) | 0.025 | 31.85 | 40.34 | 0.033 | 0.453 | 0.820 |
| **Study design** | | | | | | | | | |
| Case-control | 4 | 10 | 1.26 (0.96–1.65) | 0.095 | 9.36 | 3.81 | 0.405 | 0.513 | 0.245 |
| Cohort | 7 | 10 | 1.31 (0.96–1.78) | 0.085 | 22.34 | 59.71 | 0.008 | 0.351 | 0.721 |
| **Age** | | | | | | | | | |
| <6 years | 0 | 0 | - | - | - | - | - | - | - |
| <15 years | 10 | 16 | 1.09 (0.93–1.27) | 0.293 | 13.52 | 0.00 | 0.562 | 0.848 | 0.653 |
| All others | 1 | 4 | - | - | - | - | - | - | - |
| **Lymphomas (ICCC-3 II)** | 3 | 5 | 1.58 (0.94-2.66) | 0.083 | 4.97 | 19.55 | 0.290 | 0.457 | 0.327 |
| **Study design** | | | | | | | | | |
| Case-control | 0 | 0 | - | - | - | - | - | - | - |
| Cohort | 3 | 5 | 1.58 (0.94–2.66) | 0.083 | 4.97 | 19.55 | 0.290 | 0.457 | 0.327 |
| **Age** | | | | | | | | | |
| <6 years | 0 | 0 | - | - | - | - | - | - | - |
| <15 years | 2 | 2 | 1.30 (0.56–3.03) | 0.547 | 1.92 | 47.99 | 0.166 | - | - |
| All others | 1 | 3 | - | - | - | - | - | - | - |
| **IVF** | | | | | | | | | |
| Hematological Tumors | 3 | 10 | 1.26 (0.96–1.65) | 0.095 | 9.36 | 3.81 | 0.405 | 0.513 | 0.245 |
| Leukemias (ICCC-3 I) | 3 | 10 | 1.26 (0.96–1.65) | 0.095 | 9.36 | 3.81 | 0.405 | 0.513 | 0.245 |
| -ALL | 3 | 4 | 1.29 (0.80–2.07) | 0.291 | 3.96 | 24.15 | 0.266 | 0.007 | 0.042 |
| -Leukemia | 3 | 4 | 1.18 (0.75–1.86) | 0.483 | 4.63 | 35.26 | 0.201 | 0.245 | 0.174 |
| -non-ALL | 1 | 2 | - | - | - | - | - | - | - |

### 3.5.2. Neural Tumors

According to ICCC-3, neural tumors are grouped in the III category named "CNS nd miscellaneous intracranial and intraspinal neoplasms". MAR showed no significant association with neural tumors risk, while ART showed an increment of 21% (Table 3). Stratifying by age, a positive association between ART and neural tumors risk was found in the "all others" group (OR 1.35; 95% CI 1.02–1.77; $I^2 = 0.00\%$; $p = 0.733$; two studies). Stratifying by study design, a positive association between ART and neural tumors risk was found in the cohort studies (OR 1.21; 95% CI 1.01–1.46; $I^2 = 0.00\%$; $p = 0.543$; seven studies). No association was found between MAR, ART, or IVF and "CNS tumors".

**Table 3.** Results of stratified analysis on the cancer risk estimates in children born after MAR, ART, and IVF on the basis of type of cancer.

| Models | Studies | Effect Size | Combined Risk Estimate | | Test of Heterogeneity | | | Publication Bias | |
|---|---|---|---|---|---|---|---|---|---|
| **MAR** | N° | N° | Value (95% CI) | *p* | Q | $I^2\%$ | *p* | *p* (Egger Test) | *p* (Begg Test) |
| Neural Tumors (ICCC-3 III) | 11 | 22 | 1.05 (0.88-1.26) | 0.562 | 29.50 | 28.81 | 0.103 | 0.026 | 0.108 |
| -Astrocytomas | 3 | 3 | 1.13 (0.64–2.00) | 0.664 | 0.66 | 0.00 | 0.718 | 0.707 | 0.602 |
| -CNS tumors | 10 | 13 | 1.00 (0.80–1.24) | 0.968 | 24.03 | 50.07 | 0.020 | 0.004 | 0.028 |
| -Embryonal CNS tumors | 2 | 2 | 1.06 (0.38–2.92) | 0.916 | 1.73 | 42.26 | 0.188 | - | - |
| -Ependymomas | 1 | 1 | - | - | - | - | - | - | - |
| -Other CNS tumors | 1 | 1 | - | - | - | - | - | - | - |
| -Other glioma | 2 | 2 | 0.99 (0.30–3.28) | 0.992 | 0.05 | 0.00 | 0.826 | - | - |
| Neuroblastoma (ICCC-3 IV) | 9 | 14 | 1.21 (0.98–1.50) | 0.078 | 8.73 | 0.00 | 0.793 | 0.280 | 0.702 |
| Retinoblastoma (ICCC-3 V) | 8 | 9 | 1.49 (0.92–2.44) | 0.106 | 20.84 | 61.60 | 0.008 | 0.843 | 1.000 |
| Renal Tumors (ICCC-3 VI) | 8 | 8 | 1.22 (0.79–1.88) | 0.367 | 13.09 | 46.51 | 0.070 | 0.686 | 0.322 |
| -Nephroblastoma | 1 | 1 | - | - | - | - | - | - | - |
| -Renal tumors | 6 | 6 | 1.18 (0.64–2.17) | 0.601 | 12.66 | 60.51 | 0.027 | 0.753 | 0.348 |
| -Wilms tumors | 1 | 1 | - | - | - | - | - | - | - |
| Hepatic Tumors (ICCC-3 VII) | 7 | 11 | 2.77 (1.72–4.49) | <0.0001 | 19.85 | 49.61 | 0.031 | 0.044 | 0.102 |
| Hepatic Tumors (ICCC-3 VII) excluding Puumala et al., 2012 | 6 | 9 | 3.59 (2.31–5.57) | <0.0001 | 9.11 | 12.19 | 0.333 | 0.434 | 0.404 |
| -Hepatoblastoma | 4 | 6 | 3.03 (1.31–6.99) | 0.009 | 16.05 | 68.85 | 0.007 | 0.171 | 0.348 |
| -Hepatic tumors | 5 | 5 | 2.63 (1.60–4.31) | 0.0001 | 3.36 | 0.00 | 0.500 | 0.415 | 0.327 |
| Bone Tumors (ICCC-3 VIII) | 5 | 8 | 1.50 (0.92–2.46) | 0.105 | 15.20 | 53.94 | 0.034 | 0.508 | 0.216 |
| -Bone tumors | 5 | 6 | 1.28 (0.70–2.33) | 0.422 | 13.75 | 63.64 | 0.017 | 0.297 | 0.188 |
| -Ewing's sarcoma | 1 | 1 | - | - | - | - | - | - | - |
| -Osteosarcoma | 1 | 1 | - | - | - | - | - | - | - |
| Sarcomas (ICCC-3 IX) | 6 | 10 | 1.54 (1.18–2.02) | 0.002 | 9.46 | 4.90 | 0.396 | 0.523 | 0.325 |
| -Mesothelium and connective tissue | 1 | 1 | - | - | - | - | - | - | - |
| -Other sarcomas | 1 | 1 | - | - | - | - | - | - | - |
| -Rhabdomyosarcoma | 1 | 3 | - | - | - | - | - | - | - |
| -Soft tissue sarcomas | 5 | 5 | 1.14 (0.81–1.60) | 0.441 | 0.21 | 0.00 | 0.995 | 0.766 | 0.624 |
| Germ Cell Tumors (ICCC-3 X) | 4 | 5 | 1.13 (0.76–1.67) | 0.541 | 1.22 | 0.00 | 0.875 | 0.329 | 0.327 |

**Table 3.** *Cont.*

| Models | Studies | Effect Size | Combined Risk Estimate | | Test of Heterogeneity | | | Publication Bias | |
|---|---|---|---|---|---|---|---|---|---|
| **MAR** | N° | N° | Value (95% CI) | *p* | Q | I²% | *p* | *p* (Egger Test) | *p* (Begg Test) |
| -GCT | 3 | 3 | 0.92 (0.47–1.82) | 0.819 | 0.61 | 0.00 | 0.736 | 0.262 | 0.117 |
| -Gonadal tumors | 2 | 2 | 1.24 (0.77–2.00) | 0.366 | 0.11 | 0.00 | 0.742 | - | - |
| **ART** | | | | | | | | | |
| Neural Tumors (ICCC-3 III) | 7 | 11 | 1.21 (1.01–1.46) | 0.040 | 8.88 | 0.00 | 0.543 | 0.188 | 0.312 |
| -Astrocytomas | 2 | 2 | 1.17 (0.65–2.10) | 0.609 | 0.54 | 0.00 | 0.461 | - | - |
| -CNS tumors | 6 | 6 | 1.17 (0.90–1.52) | 0.245 | 7.11 | 29.71 | 0.212 | 0.192 | 0.188 |
| -Embryonal CNS tumors | 1 | 1 | - | - | - | - | - | - | - |
| -Other CNS tumors | 1 | 1 | - | - | - | - | - | - | - |
| -Other glioma | 1 | 1 | - | - | - | - | - | - | - |
| Neuroblastoma (ICCC-3 IV) | 6 | 6 | 1.13 (0.81–1.58) | 0.477 | 2.57 | 0.00 | 0.766 | 0.193 | 0.573 |
| Retinoblastoma (ICCC-3 V) | 7 | 7 | 1.65 (0.83–3.27) | 0.154 | 19.14 | 68.65 | 0.004 | 0.903 | 0.881 |
| Renal Tumors (ICCC-3 VI) | 6 | 6 | 1.30 (0.67–2.50) | 0.440 | 12.46 | 59.89 | 0.029 | 0.871 | 0.573 |
| -Renal tumors | 5 | 5 | 1.26 (0.55–2.89) | 0.588 | 12.30 | 67.47 | 0.015 | 0.882 | 0.624 |
| -Wilms tumors | 1 | 1 | - | - | - | - | - | - | - |
| Hepatic Tumors (ICCC-3 VII) | 5 | 8 | 3.14 (1.95–5.06) | <0.0001 | 8.25 | 15.15 | 0.311 | 0.540 | 0.458 |
| -Hepatoblastoma | 3 | 4 | 3.31 (1.28–8.57) | 0.014 | 6.01 | 50.07 | 0.111 | 0.744 | 0.497 |
| -Hepatic tumors | 4 | 4 | 3.18 (1.73–5.83) | 0.0002 | 2.22 | 0.00 | 0.528 | 0.751 | 1.000 |
| Bone Tumors (ICCC-3 VIII) | 3 | 5 | 1.86 (0.93–3.69) | 0.078 | 9.99 | 59.97 | 0.041 | 0.650 | 0.142 |
| -Bone tumors | 3 | 3 | 1.46 (0.48–4.51) | 0.507 | 9.54 | 79.03 | 0.008 | 0.619 | 0.117 |
| -Ewing's sarcoma | 1 | 1 | - | - | - | - | - | - | - |
| -Osteosarcoma | 1 | 1 | - | - | - | - | - | - | - |
| Sarcomas (ICCC-3 IX) | 4 | 7 | 1.92 (1.34–2.74) | 0.0003 | 5.28 | 0.00 | 0.508 | 0.693 | 0.881 |
| -Other sarcomas | 1 | 1 | - | - | - | - | - | - | - |
| -Rhabdomyosarcoma | 1 | 3 | - | - | - | - | - | - | - |
| -Soft tissue sarcomas | 3 | 3 | 1.25 (0.71–2.21) | 0.436 | 0.05 | 0.00 | 0.976 | 0.569 | 0.602 |
| Germ Cell Tumors (ICCC-3 X) | 2 | 2 | 0.98 (0.41–2.31) | 0.595 | 0.57 | 0.00 | 0.451 | - | - |
| **IVF** | | | | | | | | | |
| Neuroblastoma (ICCC-3 IV) | 1 | 1 | - | - | - | - | - | - | - |
| Retinoblastoma (ICCC-3 V) | 2 | 2 | 1.83 (1.00–3.35) | 0.049 | 1.17 | 14.50 | 0.279 | - | - |
| Hepatic Tumors (ICCC-3 VII) | 1 | 1 | - | - | - | - | - | - | - |
| -Hepatoblastoma | 1 | 1 | - | - | - | - | - | - | - |
| Sarcomas (ICCC-3 IX) | 1 | 1 | - | - | - | - | - | - | - |
| -Rhabdomyosarcoma | 1 | 1 | - | - | - | - | - | - | - |

In the overall analysis for neural tumors, the heterogeneity was significantly high for CNS tumors and MAR (I² = 50.02%, *p* = 0.020).

### 3.5.3. Neuroblastoma

We observed no significant association of neuroblastoma with MAR or ART. The stratified analysis showed in the "all others" group a significantly increased risk for neuroblastoma (OR 1.44; 95% CI 1.03–2.01; $I^2$ = 0.00%; $p$ = 0.879; two studies) in association with MAR. Due to the paucity of records, no analysis on the association between IVF and neuroblastoma was possible.

### 3.5.4. Retinoblastoma

MAR and ART showed no significant association. IVF significantly increased the risk of retinoblastoma (83%) at the limit of significance, although the analysis was performed on two records only (Table 3). The heterogeneity in retinoblastoma analysis was significantly high both for MAR ($I^2$ = 61.60%, $p$ = 0.008) and ART ($I^2$ = 68.65%, $p$ = 0.004).

### 3.5.5. Renal Tumors

MAR and ART did not significantly increase the risk of renal tumors. No data were available for IVF analysis (Table 3). The heterogeneity in renal tumors (VI category of ICCC-3) analysis was significantly high for ART ($I^2$ = 59.89%, $p$ = 0.0004). In the analysis for "renal tumors" (the outcome subcategory) the heterogeneity was significantly high both for MAR ($I^2$ = 60.51%, $p$ = 0.027) and ART ($I^2$ = 67.47%, $p$ = 0.015).

### 3.5.6. Hepatic Tumors

In the case of hepatic tumors (VII category of ICCC-3), both MAR and ART robustly increased the risk by 177% and 214%, respectively (Table 3). Similar results occurred in the stratified analyses considering "hepatoblastoma" and "hepatic carcinomas". Stratifying by study design, cohort studies showed a stronger association of hepatic tumors with MAR (OR 3.59; 95% CI 2.31–5.57; $I^2$ = 12.19%; $p$ = 0.333; six studies) and ART (OR 3.85; 95% CI 2.37–6.2; $I^2$ = 0.00%; $p$ = 0.586; four studies). Stratifying by age, in the "<15 years" group the risk of hepatic tumors was significantly increased by MAR (OR 3.20; 95% CI 1.76–5.80; $I^2$ = 58.49%; $p$ = 0.013; five studies) and ART (OR 2.05; 95% CI 1.39–3.02; $I^2$ = 0.00%; $p$ = 0.475; three studies).

The heterogeneity was moderate for the association between MAR and hepatic tumors ($I^2$ = 49.61%, $p$ = 0.031), while it was significantly high for MAR in the "<15 years" group ($I^2$ = 58.49%, $p$ = 0.013) and for MAR and "hepatoblastoma" ($I^2$ = 68.85%, $p$ = 0.007).

### 3.5.7. Bone Tumors

MAR and ART did not significantly increase the risk for "bone tumors" (Table 3). No data were available for IVF analysis. The heterogeneity in bone tumors (VIII category of ICCC-3) analysis wassignificantly high both for MAR ($I^2$ = 53.94%, $p$ = 0.034) and ART ($I^2$ = 59.97%, $p$ = 0.041). In the analysis for "bone tumors" (the outcome subcategory), the heterogeneity was significantly high both for MAR ($I^2$ = 63.64%, $p$ = 0.017) and ART ($I^2$ = 79.03%, $p$ = 0.008).

### 3.5.8. Soft Tissue and Other Extraosseous Sarcomas

MAR and ART increased the risk of sarcomas by 54% and 92%, respectively (Table 3). Stratifying by age, a significantly higher risk was observed in association with MAR (OR 1.81; 95% CI 1.26–2.61; $I^2$ = 7.47%; $p$ = 0.371; four studies) and ART (OR 2.05; 95% CI 1.39–3.02; $I^2$ = 0.00%; $p$ = 0.475; three studies) in the "<15 years" group. Stratifying by study design, a significant association was found for MAR (OR 1.61; 95% CI 1.20–2.17; $I^2$ = 8.13%; $p$ = 0.367; five studies) and ART (OR 1.92; 95% CI 1.34–2.74; $I^2$ = 0.00%; $p$ = 0.508; four studies) in cohort studies. No significant association was found for "soft tissue sarcomas" (Table 3).

### 3.5.9. Germ Cell Tumors

MAR and ART did not significantly increase the risk of germ cell tumors (Table 3). There were no records reporting IVF and not enough data to stratify by age or study design.

### 3.6. Publication Bias and Sensitivity Analyses

The overall analysis revealed a significant publication bias in a few cases, as shown in Tables 1–3. In particular, the Egger's test showed publication bias for MAR with neural tumors ($p = 0.026$) and hepatic cancers ($p = 0.044$). Whereas the Egger's test showed publication bias for the association of ART with all cancer sites ($p = 0.003$). The Begg and Mazumdar method detected significant bias for the analysis of association between MAR and lymphomas ($p = 0.024$). To investigate the influence of each single study on these publication biases, sensitivity analyses were carried out and the results were reported in Tables 1–3.

The sensitivity analyses showed that the estimation for some combinations of outcome and exposure were strongly influenced by any single study. For example, the effect of MAR on AML ranged from 1.29 (95% CI 0.91–1.82; $p = 0.149$) omitting the study by Reigstad et al. [19], to 1.54 (95% CI 1.10–2.16; $p = 0.012$) omitting the study by Ajrouche et al. [45]; the effect of ART on neural tumors ranged from 1.12 (95% CI 0.90–1.40; $p = 0.294$) omitting the study by Wennerholm et al. [53], to 1.31 (95% CI 1.07–1.61; $p = 0.009$) omitting the study by Williams et al. [43]; and the effect of MAR on all cancer sites in case-control studies ranged from 1.04 (95% CI 0.94–1.15; $p = 0.453$) omitting the study by Rudant et al. [42] to 1.45 (95% CI 1.31–1.61; $p = 0.001$) omitting the study by Ajrouche et al. [45].

## 4. Discussion

Considering the increasing use of MAR, the evaluation of short-term and long-term adverse health effects represents a fundamental issue in public health [1,2]. Several studies suggest that childhood cancer could represent a possible adverse effect of MAR [3]. In this meta-analysis, we investigated the possible association of fertility treatments and childhood cancer, and we found an increased risk of all cancers in children conceived by MAR, ART, and IVF. In particular, we observed an increased risk of hematological cancers, leukemias, sarcomas, and hepatic cancers in children conceived by MAR and ART.

We detected a large heterogeneity when analyzing the possible association of MAR and ART with hematological cancers, leukemias, and all cancers. The large heterogeneity observed in the analysis for all cancers could be mostly attributable to the inclusion of widely different types of tumors, such as hepatoblastomas, nephroblastoma, bone tumors, sarcomas, and germ cell tumors. The heterogeneity in the high range observed for hematological cancers and leukemias is probably due to the variability of the fertility treatments included in MAR, as suggested by the stratified analysis for ART, which showed a significant heterogeneity in the moderate range.

Similar to our study, a previous meta-analysis found an increased risk of all cancers, hematological cancers, leukemias, and central nervous system/neural cancers for children conceived by fertility treatment [13]. In addition, our meta-analysis showed an increased risk of sarcomas and hepatoblastomas. By contrast, we found no significant association with neuroblastomas, retinoblastomas, and other solid cancers. Moreover, we performed ameta-analysis categorizing the outcomes according to the ICCC-3 classification system, investigating separately hepatoblastomas, nephroblastoma, bone tumors, sarcomas, and germ cell tumors, which were grouped in the "other solid cancers" category in the previous meta-analysis.

These discrepancies are likely due to the inclusion of more recent studies and to the increased length of follow up, which permitted us to improve the precision and reliability of our risk estimates. Indeed, the meta-analysis by Hargreave et al. [13] was published ahead of three large cohort studies [17,19,48], all of which are included in the present study.

In the stratified analysis by study design, each fertility treatment showed higher effects in cohort studies than in case-control studies. As the cohort studies are the most represented and the most recent studies in our meta-analysis, we can assume that the better quality of these studies (mean quality score 7.7) led to an improved precision of our results.

Stratifying by age, we found no significant association between fertility treatments and many cancers in children aged less than six years, which could be attributable to the paucity of data. In children aged less than 15 years, the stratification produced results not statistically significant for the association of hematological tumors with MAR or ART, leukemias with MAR or ART, AML with MAR, and neural tumors with ART. Further large population-based cohort studies are needed for more accurate evidence explaining the results observed for children aged less than 15 years, particularly for hematological and neural tumors, and to investigate the possible association of fertility treatments and cancer in children aged less than six years.

According to the literature [5,54], the risk of pregnancy and neonatal birth adverse outcomes is increased by factors associated with ART procedures, such as the medications used to induce ovulation or to maintain the pregnancy in the early stages, the culture media composition, the length of time in culture, the freezing and thawing of embryos, the potential for polyspermic fertilization, the delayed fertilization of the oocyte, altered hormonal environment at the time of implantation, and the manipulation of gametes and embryos.

These factors, or a combination of them, could influence the risk of cancer in children conceived after fertility treatment. The epigenetic DNA modifications and the gene expression level within the periconception period and during assisted reproduction may influence embryo development and long-term health [55]. Indeed, in ART cycles, gametes and embryos are subjected to artificial culture media, which alters the hormonal and chemical environment with potential consequences such as alterations in DNA methylation, mRNA-mediated abnormal expression of genes, and epigenetic modifications [56].

However, it should be considered that subfertile couples may have a greater risk of preexisting methylation defects and consequent imprinting disorders in their offspring [57–59]. Dysregulated imprinting activity and changes in DNA methylation profiles are common features of the development of several human diseases, such as cancer [60]. Therefore, it is difficult to evaluate whether the parental subfertility status or the ART procedures have a stronger influence on the adverse effects.

A recent meta-analysis by Lazaraviciute et al. [61] showed a higher risk of any imprinting disorder in children conceived through ART, but no evidence of generalized changes in DNA methylation of selected genes was observed. Thus, the ART related increase in cancer risk may be due to imprinting disorders not associated to DNA methylation.

During our work, the meta-analysis by Wang et al. [62] was published in June 2019. The leading differences between the study by Wang et al. [62] and our meta-analysis are represented by search strategy and inclusion criteria. We excluded articles not reporting at least one case of the outcome of interest and the risk estimate for cancer in the offspring. In particular, we excluded the studies by Bradbury et al. [63], by Källén et al. [64], by Lerner-Geva et al. [65], by Lidegaard et al. [66], and by Pinborg et al. [67], which were included in the meta-analysis by Wang et al. [62]. By contrast, we included the studies by Brinton et al. [51], by Hargreave et al. [44], by Marees et al. [36], by Munzer et al. [35], by Wennerholm et al. [53], by Spaan et al. [47], and by Spector et al. [46]. Therefore, the different search strategies and inclusion criteria yielded to dissimilar results.

The findings by Wang et al. [62] showed that children conceived by MAR had a significantly higher risk of developing overall cancer (RR 1.16, 95% CI 1.01–1.32), hematological malignancies (RR 1.39, 95% CI 1.21–1.60), leukemia (RR 1.31, 95% CI 1.09–1.57), and hepatic tumors (RR 2.26, 95% CI 1.32–3.85). In our meta-analysis, we found that MAR significantly increased the risk of developing overall cancer (OR 1.21, 95% CI 1.14–1.28), hematological malignancies (OR 1.21, 95% CI 1.07–1.36), leukemia (OR 1.17, 95% CI 1.03–1.34), and hepatic tumors (OR 2.77, 95% CI 1.72–4.49) in the offspring. Furthermore, we stratified the analyses and organized the results according to the ICCC-3, while Wang et al. [62]

analyzed a different combination of solid tumors. For example, we estimated separately the risk for "bone tumors" and "soft tissue and other extraosseus sarcomas". Moreover, we further stratified the grouped cancers in subcategories, such as leukemia/lymphoma, neural tumors, and hepatic tumors. Finally, we explored the potential relationship between different medical reproductive techniques (MAR, ART, and IVF) and stratified not only for study design but also for age.

*Study Strenghts and Limitations*

A strength of our study is that recall bias did not substantially affect our estimates. In the case-control studies, potential recall biases are common, since parents of children with cancer could be more likely to recall periconceptional events than parents of controls, leading to a possible overestimation of cancer risk. However, no overestimation happened in our meta-analysis since the stratification by study design showed higher risk estimates for cohort studies than for case-control.

The meta-analytical approach has some limitations: it cannot eliminate potential sources of error in the included studies, which also differ considerably in quality, design, data collection, definition of the exposure, and adjustment factors. Thus, our findings should be interpreted cautiously, especially in the stratified analyses by cancer types pooling few risk estimations. Risk factors for childhood cancer are largely unknown, apart from high-dose radiation exposure, previous chemotherapy, and genetic factors [68]. Unfortunately, many included studies did not consider them as adjustment factors. Instead, the most frequent adjustment factors were the child's gender and maternal age, which has been shown to influence childhood cancer risk [68]. Some included studies adjusted for birth outcomes such as low birth weight, which is related to many types of pediatric cancer. For example, children born after MAR have consistently shown to be at higher risk for low birth weight and prematurity than children born after spontaneous conception [69,70]. Low birth weight is a potential mediating factor that can lead to an increased risk of hepatoblastoma [32,71,72]. In our meta-analysis, three out of seven studies investigating the risk of hepatoblastoma considered low birth weight as confounding factor.

In this meta-analysis, we provided no specific risk estimation for the hormonal medications administered to the sub-fertile mothers. Indeed, a possible overlap with ART procedures could occur because the mothers of children conceived by ART may have received previous hormonal medications not only as first infertility treatment option, but also as part of ART to induce ovulation and/or to maintain the pregnancy in the early stages.

Considering the wide variety of treatments implied in MAR, the multiple indications for hormonal medications (e.g., fertility treatment for conception, preparation to ART, maintenance of high-risk pregnancies), and the variety of factors related to parental subfertility status, it is difficult to assess whether childhood cancer is an adverse outcome more strongly associated with fertility treatments or infertility itself.

Hence, further studies are needed to explore the possible association of childhood cancer with parental infertility, such as studies aimed to compare children conceived without ART or any hormonal treatment by sub-fertile or infertile mothers and children conceived by fertile mothers.

## 5. Conclusions

Our meta-analysis investigated the possible association between fertility treatments and childhood cancer. MAR and ART significantly increased the overall cancer risk. We found that MAR is associated with an increased risk of hematological cancers, leukemias, sarcomas, and hepatic tumors, while ART is associated with an increased risk of hematological cancers, leukemias, sarcomas, hepatic tumors, and neural tumors. IVF was associated with an increased risk of all cancers and retinoblastoma. Despite the detection of the major sources of heterogeneity in performing the stratified analysis and the sensitivity analysis, our results should be interpreted cautiously. Considering the increasing number of children conceived by MAR, the evaluation of short-term and long-term outcomes represents an important issue in public health.

**Supplementary Materials:** The following are available online at http://www.mdpi.com/2571-8800/2/4/28/s1, Table S1: Search Strategy for the Meta-Analysis; up to 3 July 2018, Table S2: Studies' characteristics included in the meta-analysis, Table S3: Description of MAR Exposure and Types of Cancer in the Selected Studies.

**Author Contributions:** All authors have contributed significantly. Authors M.C. and G.N. provided the idea and designed the study. Authors A.O. and M.C. collected the data, wrote the article, and performed reference collection. Author R.F. analyzed the data and edited the pictures. All authors read and approved the final manuscript.

**Funding:** This research did not receive any specific grant from funding agencies in the public, commercial, or not-for-profit sectors.

**Conflicts of Interest:** The authors declare no conflict of interest.

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
