# Peer review of "Cancer Risk in Children and Young Adults (Offspring) Born after Medically Assisted Reproduction: A Systematic Review and Meta-Analysis"

_2571-8800, doi:10.3390/j2040028_

Round 1

Reviewer 1 Report

In this study, the authors performed meta-analysis (18 cohort and 15 case-18 control studies, data from PubMed, Scopus and Web of 14 Science up until June 2018.), and found that Medically Assisted Reproduction (MAR) and Assisted Reproductive Technology (ART) significantly increased the risk for certain types of tumor in children born following MAR including hematological tumors, hepatic tumors and sarcomas. MAR increased Acute Myeloid Leukemia risk and ART increased neural cancer risk. This study provides evidence for the possible long-term adverse impact of fertility treatment.

Please check line 404-407, both MAR and ART are associated with an increased risk of all cancers?

Author Response

In accordance with reviewer 1,

- we modified line 404-406

- we corrected grammar errors and punctuation marks.

Reviewer 2 Report

I do not have problems and concerns with the revised manuscript. I agree to accept the manuscript.

Author Response

In accordance with reviewer 2

We corrected grammar errors and punctuation marks.

This manuscript is a resubmission of an earlier submission. The following is a list of the peer review reports and author responses from that submission.

Round 1

Reviewer 1 Report

This is a well-conducted review and meta-analysis on the pertinent topic of cancer risk among children born after MAR. The methodology and the presentation of results are done well and the supplementary materials are also useful. It could be helpful to further elaborate on how the combined risk estimate was calculated (for example, by including the model's equation). In addition, the manuscript is generally well-written, although it requires some proofreading to correct a few minor grammatical errors (e.g. lines 155-156 and 162-163).     

Reviewer 2 Report

 Cancer risk in children and young adults - review

The topic is an important one.  I have only a few comments as I agree with the question, methodology and interpretation.

The paper needs an editorial pass as there are a number of minor errors in language.

e.g.

Line 10  Background NOT Backgrounds.

Line 28 Delete “the” before assisted

Line 32 Closer to 8 million now – please update

Line 44 delete “which”

Line 55 Add reference please

Line 79  “Extracted” not “pulled out”

Line 80  This is ambiguous considering the later discussion on sensitivity analysis.

Line 100.  OK, this is also ambiguous. You scored for quality but then ignored it? Please add that you THEN went to sensitivity analyses.

Line 158  “shown” not “show”

Line 230  “robustly” rather than “deeply”

Line 337 “fertility treatments” or MAR?

Reviewer 3 Report

Title:                      “Cancer risk in children and young adults (offspring)  born after medically assisted reproduction: a systematic review and meta-analysis”

Question:            “association between MAR and childhood cancer.”

Why:                     “to continually monitor its possible long-term adverse health effects.”

What is the hypothesis?                 No

Definitions:

·         MAR:     “We grouped the fertility treatments and procedures as MAR and ART, according to “The International Glossary on Infertility and Fertility Care, 2017” 4. MAR includes all types of fertility treatments, in particular any treatment inducing, triggering, stimulating ovulation and any ART procedure, while ART includes all interventions that involve the in vitro handling of both human oocytes and sperm or of embryos for the purpose of reproduction, such as IVF and ICSI. We considered correct to stratify for IVF and ICSI (as specified in the original articles) because they are the most common ART techniques 2. Unfortunately, we found very few data on ICSI, so we considered only IVF in the stratified analysis.”

Comment. Let us see what the exact MAR definition is according The International Glossary on Infertility and Fertility Care, 2017: “Reproduction brought about through various interventions, procedures, surgeries and technologies to treat different forms of fertility impairment and infertility. These include ovulation induction, ovarian stimulation, ovulation triggering, all ART procedures, uterine transplantation and intra-uterine, intracervical and intravaginal insemination with semen of husband/partner or donor.” Thus, every assistance in case of infertility is MAR, thus also eg using a BTC or intra-uterine, intracervical and intravaginal insemination with semen of husband/partner or donor. What are the authors looking for? Just counting? But what counting?

·         Outcome:            “The outcome of interest in our analysis was childhood cancer, classified according to the ICCC-3 20. We conducted separate meta-analyses for different cancer outcomes and the main cancer outcomes were hematological cancers, neural tumors, neuroblastoma, retinoblastoma, renal tumors, hepatic tumors, bone tumors, soft tissue and other extraosseous sarcomas, and germ cell tumors.”  Comment. Why? What is the rationale? Again, what is the hypothesis?

·         Participants:       “children and young adults.” Comment. Age limits? Why age yardstick?

·         Control:                ?

Reason for in- and exclusion of the studies: no

Quality testing of the included studies: no

I missed eg this paper: Risk of cancer in children and young adults conceived by assisted reproductive technology. Spaan M, van den Belt-Dusebout AW, van den Heuvel-Eibrink MM, Hauptmann M, Lambalk CB, Burger CW, van Leeuwen FE; OMEGA-steering group. Hum Reprod. 2019 Apr 1;34(4):740-750. doi: 10.1093/humrep/dey394. Their conclusion: Overall, ART-conceived children do not appear to have an increased risk of cancer.